# Emotional Intelligence Training: Influence of a Brief Slow-Paced Breathing Exercise on Psychophysiological Variables Linked to Emotion Regulation

**DOI:** 10.3390/ijerph18126630

**Published:** 2021-06-20

**Authors:** Min You, Sylvain Laborde, Nina Zammit, Maša Iskra, Uirassu Borges, Fabrice Dosseville, Robert S. Vaughan

**Affiliations:** 1UFR Psychologie, EA3918 CERREV, Normandie Université, 14000 Caen, France; you.min14@gmail.com; 2Department of Performance Psychology, Institute of Psychology, German Sport University, Am Sportpark Müngersdorf 6, 50937 Cologne, Germany; ninazt27@gmail.com (N.Z.); masa.iskra@gmail.com (M.I.); u.borges@dshs-koeln.de (U.B.); 3UFR STAPS, EA 4260 CESAMS, Normandie Université, 14000 Caen, France; 4Department of Health & Social Psychology, Institute of Psychology, German Sport University, 50937 Cologne, Germany; 5UMR-S 1075 COMETE, Normandie Université, 14000 Caen, France; fabrice.dosseville@unicaen.fr; 6INSERM, UMR-S 1075 COMETE, 14000 Caen, France; 7School of Education, Language, and Psychology, York St John University, York YO31 7EX, UK; r.vaughan@yorksj.ac.uk

**Keywords:** heart rate variability, parasympathetic nervous system, vagus nerve, diaphragmatic breathing, abdominal breathing

## Abstract

Designing emotional intelligence training programs requires first testing the effectiveness of techniques targeting its main dimensions. The aim of this study was to investigate the effects of a brief slow-paced breathing (SPB) exercise on psychophysiological variables linked to emotion regulation, namely cardiac vagal activity (CVA), as well as perceived stress intensity, emotional arousal, and emotional valence. A total of 61 participants completed a 5-min SPB exercise and a control condition of a 5-min rest measurement. CVA was indexed with the root mean square of successive differences (RMSSD). Participants were also asked to rate their perceived stress intensity, emotional arousal, and emotional valence. Results showed that CVA was higher during SPB in comparison to the control condition. Contrary to our hypothesis, perceived stress intensity and emotional arousal increased after SPB, and perceived emotional valence was less positive after SPB. This could be explained by experiencing dyspnea (i.e., breathing discomfort), and the need to get acclimatized to SPB. Consequently, we may conclude that although physiological benefits of SPB on CVA are immediate, training may be required in order to perceive psychological benefits.

## 1. Introduction

Athletes experience a large range of emotions during competition [1]. Emotions can be considered as short-lived psychological states, but some more stable emotional dispositions can also be identified [2,3]. One of them is emotional intelligence (EI), which reflects the way individuals deal with their own and others’ emotions [4,5]. Emotional intelligence plays an important role in sport performance [6,7], and hence its training is of high relevance for athletes. The aim of this paper is to investigate whether a brief slow-paced breathing (SPB) exercise without biofeedback could be integrated into EI training, based on its effects on psychophysiological variables linked to emotion regulation.

The theoretical foundations of EI have evolved to include its different aspects via the tripartite model [8], including the knowledge, ability, and trait levels. The knowledge level reflects what people know about emotions (e.g., knowing that SPB may help them to regulate their emotions), the ability level reflects what they can do about emotions (e.g., they can perform SPB when instructed to do so), and the trait level, reflecting the person’s usual emotional repertoire (e.g., using SPB when facing emotionally challenging situations in their daily life). In recent years, the conceptualization of EI has evolved to emotional competences, as, unlike intelligence, competences can be trained [5]. Five main competences have emerged from this conceptualization, namely emotion identification, expression, regulation, understanding, and use. The current study focuses on the competence of emotion regulation.

To date, EI has been related to sport performance, as evidenced in previous systematic [6] and meta-analytic [7] reviews. In particular, EI was found to be related to athletes’ coping strategies at the subjective [9,10,11,12,13,14] as well as at the objective physiological and hormonal levels, via cardiac vagal activity [15,16] and cortisol [17] measurements. Furthermore, EI has been associated with adaptive psychological states [18] and enhanced executive functions, such as working memory [19] and decision making [20]. Finally, some direct positive associations with sport performance have been found [21,22]. Nonetheless, despite these positive associations between EI and sport performance, one has to keep in mind that EI is one of many factors that influence sport performance [23].

Given the role of EI in sport performance, EI training [24,25,26,27] has started to receive some attention in relation to athletes [28,29,30,31,32]. Despite the positive effects of a global EI training approach combining a diversity of techniques to increase athletes’ EI, the effectiveness of individual techniques on specific emotional competences remains difficult to determine. Consequently, we argue that the development and fine-tuning of EI training programs requires testing the specific effects of separate techniques in order to enhance our understanding of their influence on emotional competences. In this paper, we focus on the influence of slow-paced breathing (SPB) on the competence of emotion regulation.

SPB is a relaxation technique in which breathing frequency is voluntarily reduced from the spontaneous rate, comprised between 12 and 20 cycles per minute (cpm) [33,34], to 6 cpm. SPB has been found to positively influence emotion regulation [35,36]. SPB without biofeedback will be considered in this study, to enable a simpler implementation in the applied field where time and resources are restricted. This refers to the performance of SPB without an external device displaying live biological signals to the participant [37]. SPB is suggested to target central and peripheral mechanisms involved in emotion regulation, via its action on the functioning of the baroreflex and pulmonary afferents, as well as by triggering oscillations in brain networks involved in emotion regulation [36,38,39,40,41,42]. The mechanism underlying these effects is suggested to be the action of SPB on the vagus nerve [39], the main nerve of the parasympathetic nervous system [43]. 

The activity of the vagus nerve regulating cardiac functioning, termed cardiac vagal activity (CVA), has been found to be positively related to emotion regulation [44]. These effects are explained theoretically by the neurovisceral integration model [45,46,47], which is based on the central autonomic network [48]. This model describes how similar brain structures are responsible for emotional, cognitive, and cardiac regulation, all of which in turn influence CVA. Importantly, CVA can be indexed non-invasively via heart rate variability (HRV), the variation in the time interval between adjacent heartbeats [49,50,51]. Among the different HRV parameters that can be calculated, the root mean square of successive differences (RMSSD) has been found to index CVA most precisely and being relatively free of respiratory influences [52]. 

Among the methods that can be used to enhance CVA [53,54], SPB has been shown to influence CVA in athletes [55,56,57,58,59]. Regarding its acute effects, although CVA appears to robustly increase during SPB, these effects seem to cease simultaneously with the termination of SPB exercises [41,55,57,58,59]. However, chronic increases in CVA can be found after long-term interventions [56]. So far, acute SPB effects have been investigated with 15-min SPB sessions. This study aimed to investigate whether a shorter SPB exercise of 5-min would also trigger an increase in CVA. 

The way individuals perceive their stress and emotions is highly relevant for emotion regulation [2,60,61,62]. However, a research gap exists regarding the potential influence of SPB on subjective psychological states related to emotion regulation in athletes. Hence, addressing subjective psychological states such as perceived stress intensity, emotional arousal, and emotional valence is of high relevance for athletes, given their influence in sport [1,11,63,64] as well as in other life domains, such as the academic [65] or the professional [66] domains. The subjective effects of SPB have received little attention so far [67,68,69,70]. On the one hand, previous research [68,69] focusing on using SPB for the treatment of panic attacks implemented clinically focused self-report instruments. On the other hand, Van Diest, Verstappen, Aubert, Widjaja, Vansteenwegen and Vlemincx [67] showed that healthy participants reported higher pleasantness and lower arousal after a 5-min SPB exercise. Nonetheless, Allen and Friedman [70] noted that SPB can provoke dyspnea (i.e., the perception of uncomfortable breathing), and showed that presenting positive pictures during SPB reduced dyspnea [71]. Given these mixed findings, our study aimed to clarify the influence of SPB on self-report variables linked to emotion regulation.

To sum up, the current study aims to address research gaps found in previous literature, by investigating the effects of a brief (5-min) SPB exercise on CVA and subjective psychological variables linked to emotion regulation, namely perceived stress, emotional arousal, and emotional valence. Based on the literature reviewed, we hypothesize that, in comparison to the control condition, the brief SPB exercise would trigger an increase in CVA. However, due to the mixed findings reported so far regarding subjective states linked to SPB, we investigated this aspect in an exploratory manner.

## 2. Materials and Methods

### 2.1. The Participants

This study was part of a larger research project investigating the effects of SPB without biofeedback in athletes. The analyses presented here have not been used or published elsewhere. Sample size determination followed recommendations for HRV research [51,72]. A total of 66 athletes were recruited as participants. Exclusion criteria were self-reported cardiovascular diseases, and other chronic diseases that might influence breathing or HRV patterns, such as asthma, diabetes, psychiatric, and neurological diseases [51]. Due to technical issues, the data of 5 participants had to be excluded, and the final sample comprised 61 athletes (*M_Age_* = 22.1, age range = 18–30 years old; 25 female; BMI: *M* = 23.21, *SD* = 2.17; Waist-to-hips ratio: *M* = 0.81, *SD* = 0.08; number of sport hours per week: *M* = 7.5 h; *SD* = 3.3). The experimental protocol was approved by the Ethics Committee of a German University (Project Identification Code 06/11/2014).

### 2.2. Material and Measures

#### 2.2.1. Cardiac Vagal Activity

HRV was measured with an ECG device (Faros 180°, Bittium, Kuopio, Finland), at a sampling rate of 500 Hz. Two disposable ECG pre-gelled electrodes (Ambu L-00-S/25, Ambu GmbH, Bad Nauheim, Germany) were used. The negative electrode was placed in the right infraclavicular fossa (just below the right clavicle) while the positive electrode was placed on the left side of the chest, below the pectoral muscle in the left anterior axillary line. From ECG recordings, RMSSD was extracted using Kubios (University of Eastern Finland, Kuopio, Finland). The ECG signal was visually inspected for artefacts and these were corrected manually when deemed necessary (<0.001% of the heartbeats), as recommended [51]. To provide an overview of the different HRV parameters [51], we also extracted the heart frequency and the standard deviation of the NN interval (SDNN) for the time-domain. For the frequency-domain (Fast Fourier Transform) low-frequency (LF; 0.04 to 0.15 Hz), high-frequency (HF: 0.15 to 0.40 Hz), and the LF/HF ratio were calculated for descriptive purposes only.

#### 2.2.2. Slow-Paced Breathing

Similar to previous research (e.g., [58,73]), SPB was conducted with a video showing a ball moving up and down at the rate of 6 cpm, based on the EZ-Air software (Thought Technology Ltd., Montreal, Canada). Participants were instructed to inhale continuously through the nose while the ball was going up, and exhale continuously through the mouth with pursed lips when the ball was going down. The video displayed a 5-min SPB exercise, with inhalation lasting 4 s and exhalation 6 s, given a longer exhalation is suggested to trigger higher increases in CVA [67,74].

#### 2.2.3. Visual Analogue Scale—Perceived stress

A visual analogue scale (VAS), consisting of a 100 mm vertical line, was used to assess perceived stress intensity. The line was anchored by the words “not stressed at all” at the extreme left of the line, and “extremely stressed” at the extreme right of the line. Participants were required to indicate their perceived stress intensity by crossing a corresponding point somewhere on the line. The value of the perceived stress intensity was represented by the distance in mm from the extreme left of the line. Previous research has implemented this scale to assess perceived stress intensity [16,75,76].

#### 2.2.4. Self-Assessment Manikin—Perceived Emotional Arousal and Perceived Emotional Valence

The self-assessment manikin [77] assesses the emotional state of an individual along two dimensions, valence and arousal (we did not include the third dimension, control, given it did not fit the aim of our study). The self-assessment manikin is a picture-oriented instrument containing five images for each of the two affective dimensions that the participant rates on a 9-point scale. Valence is depicted from negative (a frowning figure), to neutral, to positive (a smiling figure). Higher scores reflect consequently a more positive valence. Arousal is depicted ranging from low arousal (eyes closed) to high arousal (eyes wide open), with higher scores representing higher arousal. 

### 2.3. Procedure

Participants were recruited through flyers on the campus of the local university as well as via posts on social network groups linked to the local university. In line with recommendations for psychophysiological experiments involving HRV measurements [51], participants were instructed to follow their usual sleep routine the night prior to the experiment, not to consume alcohol or engage in strenuous physical activity in the previous 24 h, nor to drink or eat 2 h before taking part in the experiment. All participants gave written informed consent before participation, and were informed that they could withdraw from the study at any time without any explanation or consequence. The participants had to come to the lab once, for a within-subject design. After being welcomed to the lab, they were asked to fill out an informed consent form and a demographic questionnaire regarding variables potentially influencing HRV [51,53,78]. The ECG device was attached, and participants went through a 15-min familiarization video to get acquainted with SPB. Participants had then to go through two conditions in a counterbalanced order: (1) SPB condition: 5-min baseline, 5-min SPB, 5-min recovery; (2) control condition: 5-min baseline, 5-min rest, 5-min recovery. HRV was measured continuously throughout, and all measurements were performed with eyes opened. Participants filled out the self-report items at the end of each 5-min period. At the end of the session, the ECG device was detached, and participants were thanked and debriefed.

### 2.4. Data Analysis

HRV variables were exported from the Kubios output. Data were checked for normality and outliers. Regarding outliers, 0.01% of the cases were found to be univariate outliers (>2 SD). Their exclusion from the analyses did not change the pattern of results, therefore they were retained in the final analysis. As the RMSSD data was not normally distributed, a log-transformation was applied, as recommended for HRV research [51].

We conducted a series of repeated-measures ANOVAs, with time (baseline, intervention/rest, recovery) and condition (SPB vs. control) as independent variables, and respectively RMSSD, perceived stress intensity, perceived emotional arousal, and perceived emotional valence as dependent variables. Based on our hypotheses, we focused specifically on the time x condition interactions.

## 3. Results

Descriptive statistics are presented in Table 1 for all study variables. For RMSSD (see Figure 1), a repeated-measures ANOVA with Greenhouse-Geisser correction was conducted and showed a significant main effect of time *F*(1.456, 97.385) = 51.370, *p* < 0.001, partial η^2^ = 0.46; a significant main effect of condition *F*(1, 60) = 36.825, *p* < 0.001, partial η^2^ = 0.38, and a significant time x condition interaction effect *F*(1.709, 102.567) = 60.478, *p* < 0.001, partial η^2^ = 0.50. Nine follow-up post hoc *t*-tests were conducted, adjusting alpha level with Bonferroni correction to 0.006 (0.05/9). RMSSD was higher for in the SPB condition when participants performed SPB in comparison to the rest control measurement, *t*(60) = 12.553, Cohen’s *d* = 1.61, *p* < 0.001. In the SPB condition, RMSSD was higher during the SPB exercise, in comparison to before *t*(60) = 13.158, Cohen’s *d* = 1.69, *p* < 0.001; or after the SPB exercise, with *t*(60) = 12.722, Cohen’s *d* = 1.63, *p* < 0.001. No differences were found between the different measurement times of the control condition, nor across conditions between the measurements before and after SPB/rest control.

For perceived stress (see Figure 2), a repeated-measures ANOVA with Greenhouse-Geisser correction was conducted and showed a significant main effect of time *F*(1.676, 100.589) = 8.535, *p* < 0.001, partial η^2^ = 0.13; a significant main effect of condition *F*(1, 60) = 18.385, *p* < 0.001, partial η^2^ = 0.24, and a significant time x condition interaction effect *F*(1.506, 90.363) = 16.656, *p* < 0.001, partial η^2^ = 0.22. Nine follow-up post hoc *t*-tests were conducted, adjusting alpha level with Bonferroni correction to 0.006 (0.05/9). Perceived stress was higher in the SPB condition after SPB in comparison to after the rest control measurement, *t*(60) = 7.160, Cohen’s *d* = 0.92, *p* < 0.001. In the SPB condition, perceived stress was higher after the SPB exercise, in comparison to after pre-rest *t*(60) = 5.343, Cohen’s *d* = 0.69, *p* < 0.001; or post-rest, with *t*(60) = 6.172, Cohen’s *d* = 0.79, *p* < 0.001. No differences were found between the different measurement times of the control condition, nor across conditions between the measurements before and after SPB/rest control.

For perceived emotional arousal (see Figure 3), a repeated-measures ANOVA with Greenhouse-Geisser correction was conducted and showed a significant main effect of time *F*(1.818, 109.065) = 7.735, *p* < 0.001, partial η^2^ = 0.11; a significant main effect of condition *F*(1, 60) = 9.421, *p* = 0.004, partial η^2^ = 0.13, and a significant time x condition interaction effect *F*(1.559, 93.519) = 19.604, *p* < 0.001, partial η^2^ = 0.25. Nine follow-up post hoc *t*-tests were conducted, adjusting alpha level with Bonferroni correction to 0.006 (.05/9). Perceived emotional arousal was higher in the SPB condition after SPB in comparison to the control condition after the rest control measurement, *t*(60) = 6.798, Cohen’s *d* = 0.87, *p* < 0.001. In the SPB condition, perceived emotional arousal was higher after the SPB exercise, in comparison to after pre-rest *t*(60) = 5.557, Cohen’s *d* = 0.71, *p* < 0.001; or post-rest, with *t*(60) = 5.233, Cohen’s *d* = 0.67, *p* < 0.001. No differences were found between the different measurement times of the control condition, nor across conditions between the measurements before and after SPB/rest control.

For perceived emotional valence (see Figure 4), a repeated-measures ANOVA with Greenhouse-Geisser correction was conducted and showed a significant main effect of time *F*(1.456, 87.379) = 3.669, *p* = 0.043, partial η^2^ = 0.06; no significant main effect of condition *F*(1, 60) = 0.973, *p* = 0.328, partial η^2^ = 0.02, and a significant time x condition interaction effect *F*(1.957, 117.439) = 13.953, *p* < 0.001, partial η^2^ = 0.19. Nine follow-up post hoc *t*-tests were conducted, adjusting alpha level with Bonferroni correction to 0.006 (0.05/9). Perceived emotional valence was lower in the SPB condition after SPB in comparison to the control condition after the rest control measurement, *t*(60) = 4.333, Cohen’s *d* = 0.56, *p* < 0.001. In the SPB condition, perceived emotional valence was lower after the SPB exercise, in comparison to after pre-rest *t*(60) = 4.670, Cohen’s *d* = 0.60, *p* < 0.001; or post-rest, with *t*(60) = 3.816, Cohen’s *d* = 0.49, *p* = 0.003. No differences were found between the different measurement times of the control condition, nor across conditions between the measurements before and after SPB/rest control.

## 4. Discussion

The aim of this study was to investigate the effects of a brief SPB exercise on CVA and subjective psychophysiological variables, to understand its potential role within EI training programs. Our hypothesis regarding CVA was confirmed, given an increase was found in comparison to the control condition. Regarding the subjective variables, findings showed that the 5-min SPB was perceived as more stressful, with higher emotional arousal, and with more negative valence than the resting control condition. These findings are explained below in accordance with current literature in the field.

Regarding CVA, our findings are in line with previous literature, given so far a robust increase in CVA has been systematically demonstrated during SPB, while returning to levels close to baseline when ceasing the SPB exercise [55,57,58,59]. This acute increase may occur due to the action of SPB on the baroreflex [41] and on pulmonary afferents [38], thus stimulating the vagus nerve [39,42]. The novelty of the current findings resides in the fact that previously the increase in CVA during SPB had been mostly documented with longer SPB practice times of 15-min [55,57,58,59], while a 5-min SPB exercise was used in the current study. The fact that CVA returns to baseline after SPB would reflect the cessation of vagus nerve stimulation. However, increases in CVA found at rest after long-term SPB interventions (30 days, 15-min per day) would suggest that the regular stimulation of the vagus nerve via SPB leads to chronic increases in CVA. The underlying mechanisms still need to be uncovered, but we can speculate at this stage that those announced earlier, involving the functioning of the baroreflex, of pulmonary afferents, and the effects on brain networks involved in emotion regulation may certainly play a role in these chronic adaptations [36,38,39,40,41,42]

Our results regarding the subjective experience revealed that the 5-min SPB exercise was perceived as more stressful, with higher emotional arousal, and with more negative valence than the resting control condition. These results appear contradictory to those of Van Diest, Verstappen, Aubert, Widjaja, Vansteenwegen and Vlemincx [67], who found that a similar 5-min SPB exercise, with an approaching inhalation/exhalation ratio (3 s inhalation and 7 s exhalation, in comparison to 4 s inhalation and 6 s exhalation in our study), decreased perceived emotional arousal and increased perceived emotional valence in comparison to a baseline condition with spontaneous breathing, while they did not report any change in perceived stress intensity. These results are quite surprising, given the same instrument, the self-assessment manikin [77], was used to assess perceived emotional valence and perceived emotional arousal. We may speculate that in our study participants might have been experiencing dyspnea and found the breathing uncomfortable, potentially due to hyperventilation or unusual additional strain on the respiratory muscles. Previous research has documented dyspnea in individuals performing SPB [70,79], and have suggested the use of positive emotion induction (e.g., pictures linked to positive emotions) to counteract the potential discomfort triggered by SPB. Specifically, Allen and Friedman [70] instructed participants to inhale when a black screen appeared, and exhale when the positive pictures appeared, consequently linking positive pictures to the activation of the parasympathetic nervous system. The control condition did not include the positive pictures, but only displayed HRV-biofeedback. The authors found that dyspnea was rated as less unpleasant and less intense with positive pictures compared to the condition with HRV-biofeedback. Consequently, future research may consider using this strategy to decrease potential dyspnea, and control more strictly that participants adopt a shallow breathing technique, to avoid hyperventilating and decrease the solicitation on the respiratory muscles. Finally, future research designs may consider testing the effects of multi-sessions interventions, given the repeated realization of slow-paced breathing appears to improve participants’ experience [80]. The use of relaxing music or sounds may also help to improve participants’ experience.

Our study had some strengths, but also some limitations. Future studies should investigate whether participants are hyperventilating by using a capnometer to measure end-tidal carbon dioxide (e.g., [81]). Increased tidal volume may also have triggered more strain on respiratory muscles, and this should be controlled in further research. Additionally, given different sports contribute differently to CVA [82], future research should consider the type of sport practiced by athletes when interpreting the results of slow-paced breathing. Finally, it should be noted that we focused here on SPB without using a biofeedback device. The use of SPB has so far mostly been realized with biofeedback, showing associations with a large range of positive physical and mental health outcomes [83]. In sports, SPB with biofeedback has been associated with improved performance [84,85]. However, given biofeedback requires the use of additional devices to display the live biological signal, it is less accessible to people from lower socioeconomic backgrounds and in developing countries. Importantly, so far there is no evidence that SPB performed with biofeedback induces different physiological (CVA) or subjective (state anxiety) changes than when realized without [86]. Our results show that positive physiological benefits can be achieved using SPB without biofeedback, as indicated by increases in CVA. However further research should investigate whether the subjective experience of the participant differs between both modalities.

## 5. Conclusions

To conclude, the aim of this study was to investigate the use of SPB without biofeedback on CVA and self-report parameters linked to emotion regulation, to understand its unique added value within EI training programs. Findings showed that while CVA increased in the SPB condition, self-report measures displayed a more contrasted effect for emotion regulation, with increased perceived stress and emotional arousal, and more negative emotional valence. Altogether, these findings suggest that SPB had a positive influence at the physiological level; however, participants may have experienced some discomfort implementing this breathing technique. Consequently, future research and applied implementation of SPB should consider these findings by instructing participants not to hyperventilate, and potentially adding positive pictures to the SPB exercise [70,79]. 

Despite contrasting findings related to the subjective experience, findings regarding CVA are particularly relevant to consider, due to its role in emotion regulation and overall self-regulation [46,51,87,88,89,90,91]. CVA appears consequently as a relevant physiological marker to index the effectiveness of techniques used in EI training programs. In athletes, higher CVA was found to be associated with better executive functioning and coping under pressure [16,92,93,94,95,96,97]. Given SPB has been shown to improve both CVA and executive functioning in athletes [55,57,59], it appears to be a promising technique for the applied field. Despite the existence of other techniques which stimulate the vagus nerve without requiring the active attention of the individual, such as transcutaneous vagus nerve stimulation [98,99,100], the advantage of SPB is that it does not require any device, representing a suitable low-cost, low-technology, and non-pharmacological way to stimulate the vagus nerve.

Among the techniques already integrated into EI training that target the activity of the autonomic nervous system, SPB appears consequently to be a suitable candidate for emotion regulation, even if the instructions have to guarantee that the subjective experience aligns with the physiological benefits. Consequently, based on our findings, and taking into account several aspects that may increase the subjective experience, we may recommend implementing SPB within EI training programs. Future research should investigate whether the long-term use of SPB as a performance habit [101], as previously implemented in athletes [56], could also lead to stable changes in EI, as measured with EI questionnaires [102], such as the Trait Emotional Intelligence Questionnaire [4,103] or the Profile of Emotional Competences [5].

## Figures and Tables

**Figure 1 ijerph-18-06630-f001:**
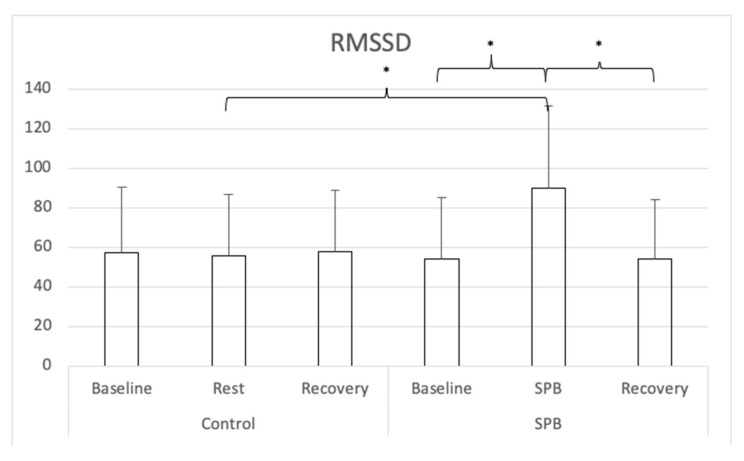
Root Mean Square of the Successive Differences. *Note:* SPB: Slow-paced breathing; * *p* < 0.006 (Bonferroni correction).

**Figure 2 ijerph-18-06630-f002:**
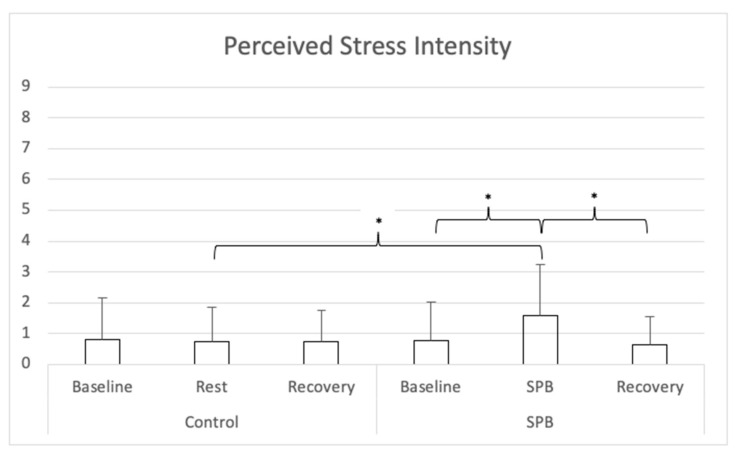
Perceived Stress Intensity. *Note:* SPB: Slow-paced breathing; * *p* < 0.006 (Bonferroni correction).

**Figure 3 ijerph-18-06630-f003:**
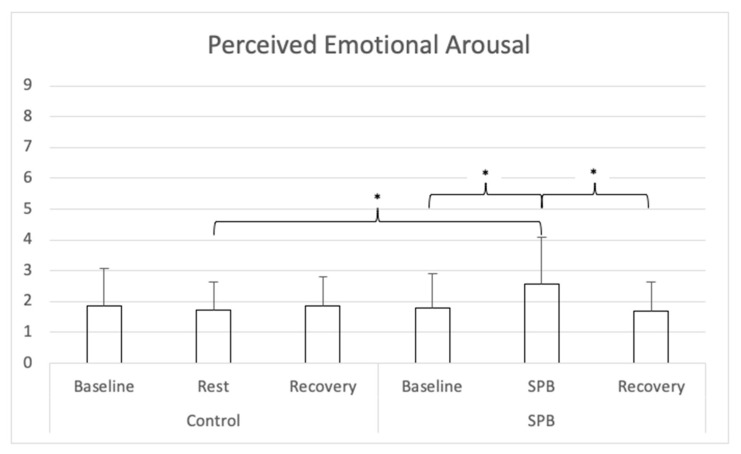
Perceived Emotional Arousal. *Note:* SPB: Slow-paced breathing; * *p* < 0.006 (Bonferroni correction).

**Figure 4 ijerph-18-06630-f004:**
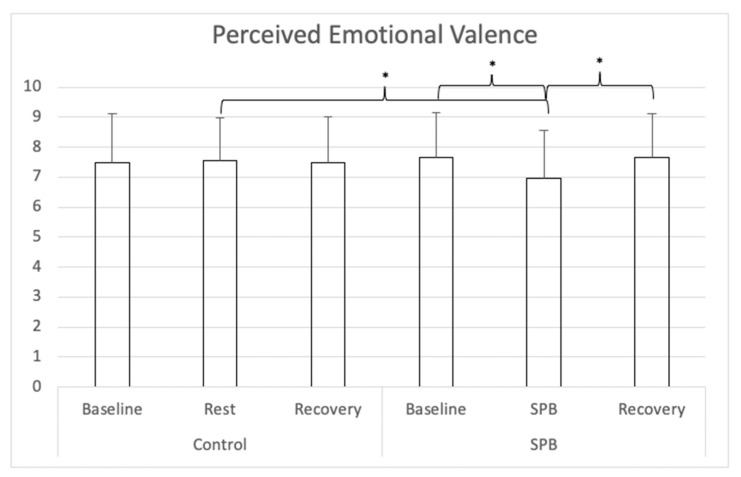
Perceived Emotional Valence. *Note:* SPB: Slow-paced breathing; * *p* < 0.006 (Bonferroni correction).

**Table 1 ijerph-18-06630-t001:** Descriptive statistics for heart rate variability.

		HR		SDNN		RMSSD		LF		HF		LF/HF	
		*M*	*SD*	*M*	*SD*	*M*	*SD*	*M*	*SD*	*M*	*SD*	*M*	*SD*
Control condition	Baseline	67.18	7.97	92.22	37.37	57.14	33.17	3735.74	6586.00	1266.25	1450.35	4.09	3.91
	Rest	67.66	7.81	91.53	34.09	55.52	30.96	3227.03	4809.28	1171.95	1315.19	4.05	3.74
	Recovery	66.71	7.76	97.36	39.60	58.08	30.77	3894.99	5905.07	1374.02	1739.26	4.65	5.22
Slow-paced breathing condition	Baseline	67.65	8.43	85.67	37.61	54.08	31.23	3185.95	4969.31	1247.59	1529.23	4.15	4.82
	Slow-paced breathing	68.59	6.86	161.10	239.21	89.93	41.39	14580.16	11277.36	1602.47	1403.40	16.13	13.15
	Recovery	67.52	8.25	88.59	35.45	53.95	30.04	2927.52	3638.94	1263.93	1869.90	4.48	5.41

Note: SDNN = standard deviation of all RR intervals, RMSSD: root mean square of the successive differences, LF = low-frequency, HF = high-frequency.

## Data Availability

Data can be shared by the corresponding author upon reasonable request.

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
