# Peer review of "Emotional Intelligence Training: Influence of a Brief Slow-Paced Breathing Exercise on Psychophysiological Variables Linked to Emotion Regulation"

_ijerph, 2021, doi:10.3390/ijerph18126630_

Round 1
Reviewer 1 Report
Thank you very much for letting me review the work.
In the manuscript, there is an excess of self-citations, which makes it difficult to have a good vision of the subject. There were no other authors who addressed the subject? this needs to be reviewed.
The use of tables that help in understanding the subject studied are missing in the results.
the introduction should relate to the discussion. appear as disconnected topics.
In conclusion, I do not believe that the article in its current state can be published.
Author Response
Reviewer 1
R1: Thank you very much for letting me review the work.
A1: We would like to thank Reviewer 1 for the taking the time to review our manuscript, and for providing helpful comments to help us improve the manuscript.
R2: In the manuscript, there is an excess of self-citations, which makes it difficult to have a good vision of the subject. There were no other authors who addressed the subject? this needs to be reviewed.
A2: We acknowledge that the manuscript contains a number of citations from our research group. This is due to the extensive research in emotional intelligence in sports conducted by our group in the past decade, as showcased in a systematic review (Laborde, Dosseville, & Allen, 2016), and with more recent work (e.g., Schütz, Rahders, Mosley, & Laborde, 2020; Vaughan, Hagyard, Brimmell, & Edwards, 2020). Still, we made sure to cite recent research from other research groups relevant to our study (e.g., Acebes-Sanchez, Granado-Peinado, & Giraldez, 2021; Castro-Sanchez, Zurita-Ortega, Ramirez-Granizo, & Ubago-Jimenez, 2020; di Fronso et al., 2020; Fernandez, Brito, Miarka, & Diaz-de-Durana, 2020; Ruiz & Robazza, 2021) to ensure a balanced perspective of the literature was provided.
Regarding slow-paced breathing without biofeedback, here again our research group has been very active in the past five years, and represents an exception, given most of the research has focused on slow-paced breathing with biofeedback, as we illustrate citing the meta-analysis of Lehrer and colleagues (Lehrer et al., 2020). Additionally, we are now preparing two meta-analyses on related topics that have been preregistered in PROSPERO, which gives us a comprehensive overview of the state-of-the-art: CRD42020200784 “Influence of breathing techniques on sport performance: a systematic review and meta-analysis”; and CRD42020173255 “Influence of slow-paced breathing on heart rate variability: a systematic review and meta-analysis“.
Nonetheless, in order to ensure that we did not miss other relevant recent research published, we reran a literature search on June 6th, 2021 in the following databases: Web of Science, Pubmed, Google Scholars; with keywords related to emotional intelligence in sports, and slow-paced breathing. No new relevant studies appeared since the first submission of our manuscript.
We hope this clarification supports an unbiased review of previous literature. However, should the reviewer wish to recommend other papers we may have overlooked, we would be happy to consider them.
R3: The use of tables that help in understanding the subject studied are missing in the results.
A3: We agree that the Results section could benefit from clearer presentation, and have now added 4 Figures to showcase our results at a glance.
R4: the introduction should relate to the discussion. appear as disconnected topics.
A4: Thank you for this comment. The role of the introduction was to set the stage regarding emotional intelligence in sports, in order to explain how a slow-paced breathing exercise would contribute to emotional intelligence training. In the discussion, we now remind this aim, in order to connect the discussion of the findings to the narrative of the introduction. We also provide a connection in the conclusion section. Regarding the discussion, given its aim to provide an interpretation of the findings, most of the narrative related to emotional intelligence found in the introduction was not deemed as crucial here, likewise this enabled us to be as parsimonious as possible. Finally, a goal of our discussion was to evaluate not just the theoretical but the applied use to the scientific community to ensure that findings are as impactful as possible.
R5: In conclusion, I do not believe that the article in its current state can be published.
A5: We would like to thank again Reviewer 1 for the helpful comments. We hope that the revision clarifies our approach to the literature, provides clearer presentation of the results in figures, and enhances the connection between the introduction and the discussion.

Reviewer 2 Report
The aim of this study is to investigate the use of brief slow-paced breathing (SPB) without biofeedback on cardiac vagal activity (CVA) and self-report parameters linked to emotion regulation.
The experiment could be improved by repeating it more often so that it would be more comfortable for the participants.
On the other hand, music or sounds could be used to make the experiment more comfortable.
There is a typo on line 260.
Author Response
Reviewer 2
Comments and Suggestions for Authors
The aim of this study is to investigate the use of brief slow-paced breathing (SPB) without biofeedback on cardiac vagal activity (CVA) and self-report parameters linked to emotion regulation.
We would like to thank Reviewer 2 for the taking the time to review our manuscript, and for providing helpful comments to help us improve the manuscript.
The experiment could be improved by repeating it more often so that it would be more comfortable for the participants. On the other hand, music or sounds could be used to make the experiment more comfortable.
This is a valid point, and we now added it to the discussion:
“Finally, future research designs may consider testing the effects of multi-sessions interventions, given the repeated realization of slow-paced breathing appears to improve participants’ experience (Szulczewski, 2019). The use of relaxing music or sounds may also help to improve participant’s experience.”
There is a typo on line 260.
Corrected, thanks for spotting it.

Reviewer 3 Report
Very interesting research, with an intersting experimental design. The introduction could be improved. The aims of the study are presented in different parts of the introduction (53-54, 70, 99-100, before a short paragraph 116-122. It would have been more comfortable for the reader, to get it in a single paragraph, after an introduction describing the State of the Art, and the pending questions.
In the Mat and Meth, the authors mention the calculation of HF, LF, LF/HF, but we do not find anymore any other information later in the paper. Why? By the way, the possible use of VLF is not discussed or mentioned.
The participants in the study, are described as athletes, but we don't have any information about the kind of sport activity. Some sports may contribute to very different CVA, a table summarizing it, would be interesting. Such information is not discussed or mentioned in the Discussion.
The Results, that are very abundant, would be easier to read, if some Tables could be proposed.
The Discussion is quite clear, and would be much easier to follow, if the results were presented in a more accessible way. Some arguments, presented in the Introduction, would be, in my opinion, more interesting, if presented in the Discussion.
Author Response
Reviewer 3
Very interesting research, with an interesting experimental design.
We would like to thank Reviewer 3 for the taking the time to review our manuscript, and for providing helpful comments to help us improve the manuscript.
The introduction could be improved. The aims of the study are presented in different parts of the introduction (53-54, 70, 99-100, before a short paragraph 116-122. It would have been more comfortable for the reader, to get it in a single paragraph, after an introduction describing the State of the Art, and the pending questions.
Thank you for this comment. We agree with the reviewer regarding the importance of the paragraph summarizing the aim of the study before the Material and Methods section, so as to provide an overview for the reader. We also considered deleting the references made to the aim of the study lines 53-54, 70, and 99-100, and asked colleagues not involved in the study to provide feedback about comprehension. They actually pointed out that they found those parts helpful to create a thread throughout the introduction, leading logically to the short paragraph lines 116-122 summarizing the aim of the study.
In the Mat and Meth, the authors mention the calculation of HF, LF, LF/HF, but we do not find anymore any other information later in the paper. Why? By the way, the possible use of VLF is not discussed or mentioned.
As indicated in the introduction, the focus on the paper is on cardiac vagal activity (CVA), the only physiological outcome that can be clearly indexed via heart rate variability measurement (Berntson et al., 1997; Laborde, Mosley, & Thayer, 2017; Malik, 1996). In this paper we decided to use the root mean square of successive differences (RMSSD) as an indicator of CVA, given it is relatively free of respiratory influences (Penttila et al., 2001). Note, we have also provided some of the other most often reported HRV parameters (e.g., SDNN, HF, LF, LF/HF), as recommended by Laborde et al. (2017), and these can be found in Table 1 for interested readers. However, given these parameters did not fit our theoretical framework based on cardiac vagal activity, they are not integrated further into our narrative. Regarding VLF, given it is suggested to reflect more long-term regulations mechanisms (Berntson et al., 1997; Laborde et al., 2017; Malik, 1996), it was not as relevant for this study investigating the effects of short-term slow-paced breathing.
The participants in the study, are described as athletes, but we don't have any information about the kind of sport activity. Some sports may contribute to very different CVA, a table summarizing it, would be interesting. Such information is not discussed or mentioned in the Discussion.
Thanks for pointing this, unfortunately we did not document the sports being practiced by our participants. We now add this as a limitation.
“Additionally, given different sports contribute differently to CVA [86], future research should consider the type of sport practiced by athletes when interpreting the results of slow-paced breathing.“
The Results, that are very abundant, would be easier to read, if some Tables could be proposed.
Thanks for your suggestion, we now added Figures to present the results in an easier and more comprehensive way.
The Discussion is quite clear, and would be much easier to follow, if the results were presented in a more accessible way.
We hope the presentation of the results now became clearer with the Figures we added.
Some arguments, presented in the Introduction, would be, in my opinion, more interesting, if presented in the Discussion.
Thank you for this comment. The role of the introduction was to set the stage regarding emotional intelligence in sports, in order to explain how a slow-paced breathing exercise would contribute to emotional intelligence training. In the discussion, we now remind this aim, in order to connect the discussion of the findings to the narrative of the introduction. We also provide a connection in the conclusion section. Regarding the discussion, given its aim is to provide an interpretation of the findings, most of the narrative related to emotional intelligence found in the introduction was not deemed as crucial here, likewise this enabled us to be as parsimonious as possible. Finally, a goal of our discussion was to evaluate not just the theoretical but the applied use to the scientific community to ensure that findings are as impactful as possible.

Round 2
Reviewer 1 Report
the authors have not sufficiently incorporated the suggested changes.
the work must be corrected.
there is an abuse of self-citations, totally inappropriate, they must be eliminated.
Author Response
We would like to thank Reviewer 1 for taking the time to read our revised manuscript and provide further comments. We acknowledge that the manuscript contains a large amount of self-citations, given as detailed in our previous response, our research group has been very active in investigating emotional intelligence, cardiac vagal activity, and slow-paced breathing without biofeedback in the past decade. That said, we made sure that we were providing an up-to-date view of the state-of-the-art, and performed a new literature search on June 6th, 2021, in the databases Web of Science, Pubmed, and Google Scholars, and no new relevant studies were found. Nonetheless, in case the reviewer would be aware of relevant research that we did not mention, we will be happy to consider his/her suggestions.
To reduce the amount of self-citations, we deleted the following ones:
Allen, M. S., & Laborde, S. (2014). The role of personality in sport and physical activity. Current Directions in Psychological Science, 23, 460-465. doi:10.1177/0963721414550705
Laborde, S., Dosseville, F., & Kinrade, N. (2014). Decision-specific reinvestment scale: an exploration of its construct validity, and association with stress and coping appraisals. Psychology of Sport & Exercise, 15, 238-245.
Laborde, S., Heuer, S., & Mosley, E. (2018). Effects of a Brief Hypnosis Relaxation Induction on Subjective Psychological States, Cardiac Vagal Activity, and Breathing Frequency. Int J Clin Exp Hypn, 66(4), 386-403. doi:10.1080/00207144.2018.1494449
Laborde, S., Strack, N., & Mosley, E. (2019). The influence of power posing on cardiac vagal activity. Acta Psychologica, 199, 102899. doi:10.1016/j.actpsy.2019.102899
Laborde, S., Eyre, J., Akpetou, J., Engler, A.-C., Hofmann, F., Klandermann, J., . . . Zajonz, P. (2020). Emotional Competences Training. In M. Ruiz & C. Robazza (Eds.), Feelings in Sport: Theory, Research, and Practical Implications for Performance and Well-being: Routledge.
Laborde, S., Mosley, E., Ackermann, S., Mrsic, A., & Dosseville, F. (in press). Emotional intelligence in sports and physical activity: an intervention focus. In K. V. Keefer, J. D. A. Parker, & D. H. Saklofske (Eds.), Handbook of Emotional Intelligence in Education: Springer.
Regarding the connection between the introduction and the discussion, we now added the following links:
“To conclude, the aim of this study was to investigate the use of SPB without biofeedback on CVA and self-report parameters linked to emotion regulation, to understand its unique added value within EI training programs. (l.341-342)”
“CVA appears consequently as a relevant physiological marker to index the effectiveness of techniques used in EI training programs.” (l.353-355)
“Consequently, based on our findings, and taking into account several aspects that may increase the subjective experience, we may recommend implementing SPB within EI training programs.“ (l.366-368)
Regarding English language, the manuscript has been proofread by our co-author Dr. Robert Vaughan, native speaker and Senior Lecturer at York St John University. Should there be any specific concerns left, thanks to let us know, and we will be happy to address them.